# ‘You’re Not Alone for China’: The First Song in Times of COVID-19 to Keep the Faith in a World Crying in Silence

**DOI:** 10.3390/bs12040088

**Published:** 2022-03-24

**Authors:** Lydia Giménez-Llort

**Affiliations:** 1Medical Psychology Unit, Department of Psychiatry and Forensic Medicine, School of Medicine, Universitat Autònoma de Barcelona, Cerdanyola del Vallès, 08193 Barcelona, Spain; lidia.gimenez@uab.cat; Tel.: +34-935812378; 2Institut de Neurociències, Universitat Autònoma de Barcelona, Cerdanyola del Vallès, 08193 Barcelona, Spain

**Keywords:** COVID-19, grief, bereavement, collective mourning, socio-ecologic model, songwriting analysis, group music therapy, resilient interventions

## Abstract

Collective mourning is an expression of societal maturity, cohesion, and respect. The world is in grief, but in early January 2020, before nobody could even imagine that SARS-CoV-2 would turn into the COVID-19 pandemic, a music video version of a pop ballad about love and isolation was spread across a Chinese social network. The song ‘You Are Not Alone’ was adapted as a cover by young foreigners living in China to express their support to bereaved families and frontline workers and encourage the people of China, their second home. At that time, the rest of the world looked to distant China but could hardly expect to face the same adversity months later. The authors reported that the music video was a spontaneous artistic expression copying such traumatic events and the mourning process. The present work analyses how the music was blended with lyrics and images describing the outbreak in Wuhan to reach their goal. The original song and this shortened version for China were compared regarding musical and lyric structures and main characteristics. Additionally, an analysis of the two videos was done regarding cinemetric variables and non-verbal communication that emphasized the power of songs to express deep sorrow and sympathy but also to give hope. Psychological first aid, the five stages of the mourning process by Kübler-Ross, the dual-process model by Stroebe and Schut, and Taylor’s tend-to-befriend provided a better understanding of the translation from interpersonal to societal mourning. Finally, other memorable songs that society spontaneously chose to be performed alone or together to cope with sudden and dramatic situations, mitigate physical distancing, and alleviate human suffering are discussed. Music, lyrics, and artistic performance are playing a key role in building social and emotional ties during this pandemic, hampering individual and social pain and sorrow despite cultural barriers.

## 1. Introduction

The world is in grief, simultaneously. However, early in January, before nobody could even imagine that SARS-CoV-2 would wind up being a COVID-19 pandemic [1], a music video by Shenzhen Meifeiya Culture Communication Co. Ltd. (Shenzhen, China), with a version of a pop ballad about love and isolation, was spread through the Shenzhen Daily official WeChat account, a social network of this Chinese city with 12.5 million inhabitants. This messaging app is the biggest social media in China, the equivalent of WhatsApp plus Facebook plus other apps, with 1.24 billion monthly active users, over one-third of who spend four hours or more on the app each day [2]. The prevalence has made WeChat an indispensable part of many people’s lives and work [2], and its effects on the subjective well-being of Chinese older adults living alone, through increased intergenerational support and social activities, have been reported [3].

The song ‘You Are Not Alone’ by Michael Jackson was adapted as a cover version by young adult foreigners living in China to express their support to bereaved families and frontline workers. They encouraged the people of China, their second home. At that time, the rest of the world watched distant China but could hardly expect to face the same adversity months later [4]. Since then, the coronavirus pandemic has disrupted the lives of millions of people worldwide. In polemology terms, it is much like total war, warfare not restricted to specific targets and resulting in massive suffering, death, and losses.

Since 31 December 2019, and as of 31 January 2022, 378 million cases of COVID-19 have been reported worldwide, and the global death toll exceeds 5.67 million [5,6]. In most cases, the impact at individual and social levels [7] is cushioned by resistance and resilience. These are natural human capacities that have been related to demographics, resources, and life stress [8], promoted by previous experiences, beliefs, and culture, among other people-centred variables and socio-contextual factors [9]. They are dynamic processes that help us first confront, then cope, and finally overcome difficulties. Hopefully, many will thrive in the face of adversity, not just adapt. However, the devastating scenarios, the confinement, and physical distancing to hamper COVID-19, along with leaving bereaved families unable to say goodbye and the frontline workers in an overwhelmed health/work system, are some of the new triggers for disenfranchised (not acknowledged as legitimate by society) and disrupted grief [10,11]. To prevent their worsening into prolonged grief symptoms and persistent complex bereavement disorder, recognition of the singularity of each personal loss and grief, better communication tools, advanced care planning, and self-care practices have been proposed [12,13]. In this context, the importance of collective mourning should be highlighted since it is considered an expression of societal maturity, cohesion, and respect [14] and should play an enhanced role in the current scenarios similar to what happens in natural disasters, wars, and other social tragedies. In the communication strategies concerning the actual risks of pandemics, it is important to impart a sense of emergency without creating social panic. However, these strategies also hamper the collective expression of grief, disenfranchise it, and force the world to cry in silence.

Through culturally shaped rituals, spirituality, arts, and symbolisms are three universal tools for physical, behavioral, cognitive, and emotional expression of individual and collective mourning and sorrow [15]. Music and songs play an important role in the grieving process, either as part of the farewell rituals or before the bereavement and anticipatory grief [16]. As neuroscience demonstrates, these roles are due to music’s extraordinary capacity to evoke feelings and recall past events, producing psychological benefits through mood regulation and empathic feelings [17]. The therapeutic value of music and songs is also investigated and proposed in healthcare, end-of-life, and life-threatening disease conditions to manage physical and psychological symptoms (i.e., [18,19]). Music and songs can also provide scenarios and tools for individual and group interventions [20,21]. Lyric analysis, improvisation, singing, and songwriting from bereaved children, preadolescents, and adolescents [22,23,24] are used in resilience programs tailored for individuals and as part of group music therapy [25]. Thematic analysis through mixed inductive and deductive content analysis methods is also used [26,27].

The present work analyses ‘You’re Not Alone for China’, the first song in the COVID-19 pandemic period performed as a music video describing the outbreak in Wuhan. The way the authors adapted the original ‘You’re Not Alone’ song, with its music blended with adapted lyrics and shocking images, is analyzed as an expression of collective mourning, similar to what is done in individual and group interventions [20,21]. The aims were to identify the key features of songwriting and imagery used by young foreigners living in China to describe their grieving experiences, the psychological constructs expressed in their process of coping with such unprecedented traumatic events, and to support bereaved families and frontline workers. Thus, the comparative analysis of the two music videos was aimed to identify the items that allowed the translation from interpersonal (original song) to societal mourning (version for China). It also considers cinemetric variables and non-verbal communication, psychological first aid, the five stages of the mourning process by Kübler-Ross, the dual-process model by Stroebe and Schut, and Taylor’s tend-to-befriend model.

## 2. Materials and Methods

‘You Are Not Alone for China’, the first cover with lyrics for this COVID-19 pandemic spread through the Shenzhen Daily official WeChat account, was analyzed and compared to the original song. For the current analysis, the access to this material was done through one of the few international open-access links offered by users that shared the video in international social networks (YouTube 10 February, MsWaiting, https://youtu.be/SbYg0i7TH4A, accessed on 10 February 2020). From here on, ‘You Are Not Alone for China’ will be referred to as ‘version’ or ‘version COVID-19′. The authors, Shenzhen Meifeiya Culture Communication Co. Ltd., edited this music video devoted to COVID-19 as an adapted version of the pop ballad by Michael Jackson ‘You Are Not Alone’ composed by Robert S. Kelly and edited by Sony/ATV Music Publishing LLC, Peermusic Publishing, 1995. For track listings of the CD single and 12” single, details are given from an international source of this song (https://en.wikipedia.org/wiki/You_Are_Not_Alone, accessed on 29 August 2020). From here on, ‘You Are Not Alone’ will be referred to in the text as ‘original’ or ‘original song’, and the ‘You Are Not Alone’ Official video, Vevo: https://youtu.be/pAyKJAtDNCw, accessed on 29 August 2020 will be used.

The original music video produced by Robert Kelly and Michael Jackson in 1995 and this shortened version for China were first compared regarding musical and lyric structures and main cinemetric characteristics. After that, the songs were analyzed in terms of non-verbal communication, social-ecological models [28,29], the phases of the mourning process described by Elisabeth Kübler-Ross [30] and Stroebe and Schut [31]. Finally, other extraordinary songs that authors and society have spontaneously chosen to be performed alone or together to cope with the sudden and dramatic situations amidst this coronavirus pandemic, mitigate physical distancing, and alleviate human suffering are discussed.

The music video clip was performed by 39 men, 23 women, 5 girls, and 3 boys, who appeared signing or playing music. That is, at least a group of 60 people (52 adults and 8 children) participated in this collective work that involved different artistic activities (namely, songwriting, singing, playing music, and video composition) similar to those used in resilient programs and group music therapy [25]. The qualitative analysis of these music video clips (the original song and the version COVID-19) was done in terms of the musical structure and lyrics regarding the social-ecological model and the grief and mourning process phases. The version for China was performed and shared as a video clip. Therefore, non-verbal communication was also analyzed and compared to the original. The music videos’ storytelling constituted the empirical data, and a hermeneutic methodological approach defined as the interpretation of stories based on understanding and preunderstanding [16] was used.

For illustration purposes, the artistic aspects ‘illuminated borderlines’ or ‘transparency’ were applied to transform the appearance of the music video’s images of the official song and its version for China (Powerpoint Version 1908, Microsoft Office 365 ProPlus).

### 2.1. Music and Non-Verbal Communication

The original song and its shortened version were compared concerning their musical structure and main characteristics. According to non-verbal communication, and for each video frame, the ‘physical characteristics’ and ‘dynamic behaviors’ were noted as follows:

Physical characteristics: Design and spatial disposition; appearance; the number of persons. Several cinemetric measures were noted, such as the average shot length, categorizing the shots by field size, camera placement, and angles. Also, the seven basic ‘field sizes shots’ were considered and rated on presence/absence. From distant to closer-sized shots: extreme long-shot, long shot, full shot, American or ¾ shot, medium shot, close-up shot, and Italian shot or extreme close-up shot. Camera placement and angles were also noted when emphasizing the shots’ meaning.

Dynamic behaviors: Gestures and body movements, eye contact, posture, body contact, spatial behavior, paralanguage, and facial expressions concerning the six basic emotions (surprise, fear, disgust, anger, happiness, and sadness) were registered as dichotomy shown/not shown.

### 2.2. Lyrics and Verbal Communication

The lyrics’ grammatical and functional analysis in the original and versioned music video was performed concerning verbal communication. Attending to the entire composition, stanzas, verses, and chorus, a hermeneutic approach was used to proxy the lyrics’ meanings. The main transformations were noted and compared to the original version.

### 2.3. Societal-Ecological Model

According to Bronfenbrenner’s bioecological model of human development, the interactions between the individual/s and the environment in a social system were evaluated [28]. Thus, a method assessing the increasing complexity levels was adapted from recent literature [32]. The six levels of complexity considered were the following: 1. Individual, 2. Microsystem or interpersonal sphere (family, neighborhood, school/university, hospital), 3. Mesosystem or organizational, 4. Exosystem (societal and public policy, industry), 5. Macrosystem (attitudes and cultural ideologies), 6. Chronosystem (historical and temporal aspects of time and its effect on an individual; includes one’s own experiences and closeness).

### 2.4. Mourning Process

The song’s verses were analyzed according to the five stages of grief and mourning described by psychiatrist Elisabeth Kübler-Ross [30]. Namely 1. Shock and denial, 2. Anger, 3. Bargaining, 4. Depression, and 5. Acceptance. However, the most contemporary theories of bereavement consider adaptative coping with the loss being composed of confrontation, avoidance of loss, and restoration stressors [33,34]. Therefore, the dual-process model of coping with bereavement by Stroebe and Schut [31] identified a dynamic, regulatory coping process of oscillation between loss- and restoration-oriented processes in everyday life experience and was also taken into account.

## 3. Results

Table 1 summarizes the comparative analysis of the infobox song ‘You Are Not Alone’ and the shorter cover video version for China elaborated by the young singers and musicians.

As shown in the timeline depicted in Figure 1, ‘You are not alone for China’ was the first song in times of COVID, before WHO declared the COVID-19 outbreak a global health emergency that spread internationally.

As Xiaochun reported in the interview with the authors [35], the video published at the Shenzhen Daily official WeChat account was a spontaneous collective artistic expression of young foreign people working at Shenzhen Meifeiya Culture Communication Co. Ltd. to cope with the traumatic events and had a significant social impact as shown by a fast spread through users of this social media. Abroad, this version was internationally spread through WhatsApp and other private social media due to copyright restrictions. Its chronological appearance can be recorded from other open-access international social media (i.e., YouTube/Facebook) that refer to 1 February, in China, and 15 February 2020, in Spain. On 30 January, the World Health Organization (WHO) declared the COVID-19 outbreak in Wuhan a public health emergency of international concern, while it was declared a pandemic on 11 March 2020 [5].

The analysis of the musical structure is illustrated in Figure 2. In the version for China, the rhyme scheme ‘AABB’ of the verses and four line stanzas was preserved, but the ballad’s musical structure was modified in its length, structure composition, and functional grammatical aspects. The ‘You Are Not Alone’ official music video (5:35 min) was reduced to 1:08 min (4:27 min duration). The instrumental intro, the two verses, the pre-chorus, and the two choruses followed the original song’s structure, but after a brief instrumental guitar solo that introduces the bridge, the subsequent original structure of Verse-Verse-(pre)-Chorus-Chorus-Bridge-extended Chorus-Ending (Repeat and Fade) was altered. Thus, the core Verse-Chorus cycle was broken in the version for China. Instead of the new cycle starting with two verses, Chorus 1 was created and reiterated three times, with a change in the background instruments (omission of the guitar) in the third repetition, followed by a fourth chorus, identical to Chorus 2. Then, an instrumental guitar solo built the song into the high intensity of the last two choruses. These choruses had a significant lyric change compared to previous ones. They retain ‘Stay strong’ (in Chinese) messages, but differ in background instruments: a guitar in Chorus 3 and a Chinese erhu violin in Chorus 4. The ending is a sudden stop.

The common parts between both versions (as shown in Figure 2) were chosen for further analysis. Table 2 summarizes the grammatical and functional analysis of the verbal communication expressed in the lyrics. After that, Figure 3, Figure 4 and Figure 5 illustrate the video frames and depict the lyrics as well as the field size shots of the original version (Figure 3) and the cover version (Figure 4 and Figure 5).

In the original video, the first images associated with the musical entrance suggest the lonely feelings of a man, looking serious or sad, looking down while walking through a path, surrounded by a line of photographers flashing their cameras. In this version, the musical entrance is encompassed by three aerial views of empty rail, river, and land transport systems of Shenzhen, a city of 12.53 million inhabitants.

The primary grammatical and functional transformation of the original song relies on the change of pronouns. The first singular person of the male gender was changed to the first plural person representing adult males and females from several nationalities as well as the local males and females of different ages. This change is a key point, as will be detailed below. The expression of an adult man’s loneliness due to losing a woman’s love (original song) was adapted to the COVID-19 situation. The version expressed wholehearted sorrow, encouragement, and support to a country in grief for the outbreak while the rest of the world was still watching in astonishment.

The video clip images, the English and Mandarin subtitles, and the author name (Shenzhen Meifeiya Culture Communication Co. Ltd.) allow us to recognize that this plural form corresponds to a young adult male and female foreigners group living and working in China. They identify themselves as foster sons/daughters in the last verse (‘And we wanna sing to our second home we’re here ’), which is done just before the start of the elongated chorus where the song is sung in Mandarin and follows in a mixed Mandarin-English manner. From the first stanza, the images alternate the lead singers with real scenarios of the pandemic in the streets, the hospital staff caring for patients, and the staff contacting their children through mobiles. The camera shot’s most significant aspect is the appearance of the extreme close-up shot, also called Italian shot, of frontline workers’ eyes and forefront. These shots imply visual contact and a shortening of interpersonal distance between the observer and the subject. Finally, the chorus starts with Chinese children and follows with teenagers and young Chinese adults. The series starts with single voice performances but follows two shots and group shots as the video progresses.

In the version for China, there are nine verses where the lyrics remain intact, 4 out of 11, if we do not consider the chorus. These verses start each of the three stanzas and refer to the grief’s temporal aspect (Stanza 1, verse 1.’Another day has gone’; Stanza 2, verse 1. ‘you never said goodbye’; Stanza 4, verse 1. “Everyday…). Concerning the mourning process, they correspond to the first stage (shock and denial).

In both songs, the first verse (‘Another day has gone’) emphasizes a loss that lasts over time. Already in the original song, the verse can be interpreted as a euphemism. It refers not only to the days lost but also to how many days the person has been gone. In the original song, this beginning is reaffirmed by other expressions of perpetual unwanted loneliness (‘I’m still alone’), of incomprehension (‘How could this be’), and how the absence manifests (‘You’re not here with me’) that leads to a need that will be the central theme of the song (‘You are not alone’). In the COVID-19 version, the loneliness concept (being alone) is not presented per se, and the second verse has been changed for an expression that talks about resistance, as it points at collective hope as a solution to achieve group strength. Thus, although one person is singing, the singer uses a first-person plural pronoun, thus betraying a collective’s name. The adaptation implies a change towards positive thinking, since the confirmation of a psychologically painful situation that remains unchanged over time, based on an existential aspect (to be) of a social nature (alone), is substituted by a confirmation of something (‘hope’) that the community (‘our’) has and could use, thus referring to the future. This spiritual element (‘hope’) is identified as a transforming agent capable of converting the group (‘can make us’) into a group with an essential virtue to resist and overcome vicissitudes (‘strong’). Therefore, this second verse corresponds to the third phase of mourning, the bargaining in a chronosystem that talks about the present (hope) and the future (strong).

The first stanza ends with the first stage of mourning, expressing the need to comprehend (‘How could this be’), as done in the original song. However, here the question is encompassed with the gesture of hands praying. It follows with images describing the incredible scenario of the silent streets in a city of 12.53 million inhabitants and a final close-up shot of the singer with the gesture of silence or index finger pointing up to call attention. The use of the verb ‘to rule’ (in ‘The silence rules the streets’) also suggests the imperative confinement. This last verse has a 360° function, as it closes the first stanza by recalling the first images of a train station, one of the rivers of Shenzhen, and a car entering an empty highway. In the original song, the images used for this last and the previous verse are opposed to verbal communication. Thus, words refer to ‘You’re not here with me’, but the images evoke the past, using a ‘two-shot’ in an ‘extreme long view’ that reproduces the intimate scene of ‘Daybreak’, the famous masterpiece painting of Mitchell Paris. The emulation of the most famous art print of the 20th century helps us know the magnitude of the loss and the sympathy/empathy feelings it might want to provoke.

In the second stanza, the opening verse (‘You never said goodbye’) is quite preserved. In both cases, the absence of an actual act (saying goodbye) to accept a person’s loss is expressed. However, in the COVID-19 version, the personal pronoun is changed from the second person singular (you) to the first person plural (we). This change gives the verse a significant change of meaning. In the original version, the speaker refers to the partner (you) as the one that omitted saying goodbye. In the version, the speaker talks using the plural (we), on behalf of everyone else, and uses the temporal verb ‘could not say’, referring to the impotence of those who could not stand by their relatives when dying from COVID-19. This statement of a feeling of sadness is confirmed with the second part of the sentence that specifies, using the direct object, ‘to those for whom we cry’.

In contrast, in the original song, the grammatical structure is changed. The second verse opens a question (‘Someone tells me why’) demanding, no matter who, a reason for an act that seems not to be yet understood (Did you have to go) and that has a significant impact as the consequence of losing affection (cold/sadness as the antagonist of warm/happiness) not only affects the individual but to his whole ecological system (my world). Version for China uses the last two verses to express the families’ emotional devastation, a sacrifice to save their beloved ones. Here, they use the first plural pronoun ‘our beloved ones’, indicating that the singers are already part of these families.

In both songs, the third stanza refers to the present, starting with ‘Every day’. However, while in the original song, the subject is inactive (I sit) and asks for clues to understand how ‘did love slop away’, the version for China refers to action (we pray) to keep that strength than in previous verses was shown as coming from hope. The prayer is also to do another action: defeat the fear, which is the emotion behind negation, the first stage of mourning. A will follows it to sing to their second home to express their support with an important statement: ‘we’re here’, that in the context of COVID-19 and migratory processes, the copying-with-stress strategy flight-to-fight is not to flight (escape, repatriation) but to stay. In contrast, the original song refers to the presence of someone that whispers to say something that will mitigate (as unveiled in the chorus) the feelings of loneliness.

The central conceptual part of the Chorus ‘You are not alone, I am here with you’ is preserved in the version for China, but the singer (now in first singular person) use the Chinese language and alternates with English in the next ones, as the instruments electric guitar, and traditional erhu violin also do in Chorus 3 and 4. The following verses refer to the past and the future, contrasting the challenging path and a future where their hearts light the way. The action of ‘working together’ as a tool that ‘can’ make it possible to ‘win this fight’ and a clear statement of the song’s intention: ‘I want you to know that I’ll be there. We’ll be there’. In the original song, the voice that whispers in the ear means that despite the distance (‘we are far apart’), she has him in her heart and a clear statement of the whispering’s intention to say ‘You’re not alone’. Still, the verses continue referring to loneliness (alone, alone) and ‘why, alone’, so there is incongruence with the previous verses, and it is not clear if the loneliness and the incomprehension have been solved.

The chorus also continues with the participation of seven Chinese children and, after that, young adults. All of them call the country (‘Zhong Guo’) and Wuhan, the focus of the epidemic, while shouting ‘Jia you’ (in English, ‘Stay strong’) and making the Chinese cultural gesture of strength. A traditional Chinese erhu violin is here the background instrument.

Finally, Figure 3 and Figure 4 illustrating the common parts of the original video, composed of 24 frames (Figure 3), and the version for China, show that now the same song structure is fragmented into 50 frames (Figure 4). Figure 5 shows the frame structure and composition for the rest of the song, composed of 98 frames. In the insets, the field size shots are depicted. As summarized in Figure 6, the frame analysis shows that the version is developed using an increased number (×2) of frames at each structural part of the song (see Figure 6A). As a total (see Figure 6B), 50 frames instead of the 24 capture the story’s intensity and speed. The field size shots comparison (Figure 6B) summarizes the enrichment of the visual storytelling with close-up shots (original: in 7 out of 24 shots; version: in 24 out of 50 shots). The Italian shot, an extreme close-up shot, and full shots are noted in the frames with more robust storytelling expression.

The comparative analysis of the version for China versus the original (see Table 3) shows that the songwriting under societal bereavement and grieving of this COVID-19 pandemic involved substantial changes in the actor, direction, goals, facts, cognition, emotions, actions, and time frames, all aspects that play a role in psychological first aids.

## 4. Discussion

In the present work, one of the first public cultural expressions of societal bereavement and grieving of this COVID-19 pandemic is analyzed. It is a music video of a pop ballad version about love and isolation arranged by young adult foreigners living in China and spread through social networks early in February 2020. The internationally awarded ballad ‘You Are Not Alone’ by the king of pop Michael Jackson was transformed into a cover version to express their support to bereaved families and frontline workers, encouraging the people of China, their second home. The songwriting and artistic (imagery) adaptations to create the version ‘You’re Not Alone for China’ are analyzed in terms of music structure and lyrics, verbal and non-verbal communication, social-ecological models, and the mourning process phases. This first song in the COVID-19 pandemic is discussed in the context of other musical expressions as a tool for social convergence in the current times.

### 4.1. You’re Not Alone

The most difficult situations in life are even more so if they are unexpected, occur unpredictably, or feel alone. So it happened in the COVID-19 pandemic outbreak [5,12]. The music video’s core theme, ‘You are not alone for China’, relied on these facts. Mainly because, as shown by the timelines, it was written and performed during the first weeks of the health crisis in Wuhan before it was aired online early in February. In some cases, the severity and impact of critical incidents make them be described as traumatic experiences due to their significant impact and long-lasting effects. Thus, it is also described in the adapted lyrics and illustrated with the images taken in real settings. In many ways, accompaniment in grief is still a taboo topic [36]. Individual and social disenfranchised grief, which is unusually emerging in this worldwide scenario [10,11], is euphemistically expressed or relieved through resilient manners such as music and other cultural expressions of the human mind [12,25,37].

As discussed above, ballads are tunes with a strophic form, comprised ordinarily of stanzas set to a rehashing melody tune. A story, frequently sentimental, historical, or tragic, is sung in a narrative style. Most popular ballads have a verse-chorus structure that alternates in cycles, with subtle (maybe only one word) or profound changes to increase impact. An inquiry and answer configuration can be incorporated with the melody, as it is done in the verses of ‘You are not alone’ that express sorrow and the first mourning phase of negation. The question asked in one verse finds its answer in the next one. The verse sets up the chorus lyrically and builds anticipation for its memorable refrain, the song’s most poetic part. In ‘You are not alone for China’, the chorus is prolonged, providing more strength to a collective message to keep the faith and stay strong.

### 4.2. Chronosystem and Socio-Ecological Perspective

The chronological appearance of the version ‘You’re Not Alone for China’ concerning the COVID-19 timeline plays a key role. It emerged in the very early moments when the Chinese people were confronting the severity of the disease, and the rest of the world was watching with astonishment and surprise. In the pandemic timeline, that would correspond to the first stage of the mourning process, according to Kübler-Ross [30] (knowing someone else misfortunate death or proximity to death), that follows with negation (difficulty to believe or accept as a real fact). This tempo also adds extra value to the singers’ attitude, representing an international community. This fact could be interpreted as a major point of strength and support since, at that time, the international community they represent was watching and had not yet publicly expressed much support. It looked at the Chinese scenario as a severe but localized problem while closed frontiers to protect itself.

Here, the author proposes that time is the key factor in the COVID-19 pandemic, as the impact of the disease in societies, even in those with the best healthcare systems, could be referred to by the confluence of seven Ts. Namely, 1. the fast-spreading of this highly infectious disease, 2. the fast-speed of mutation of the virus into new variants, 3. the silenced time, while asymptomatic people can infect others, 4. the lack of time for healthcare systems to hamper the exponential number of people and urgent-decision fatigue, 5. the long-care period in intensive care units, 6. the long-lasting and complex recovery of survivors and society, 7. the lack of time for families to say the final goodbye, which increases the risk for prolonged disrupted or pathological grief. Here, this seventh-time dimension is important to note since it has been reported that the COVID-19 pandemic is robbing families of the chance to say a final goodbye. Silent cries intensify the pain of the life that is ending [38], mainly in those cultures where the family healing process is strictly linked with performing goodbye rituals. These questions have strong implications in the mourning process and well-being [34,38], with variations according to the different cultural and traditional rituals across the entire world [15], which have deserved a number of international scientific reports since the beginning of the pandemic [34,36,38,39,40,41,42,43,44,45]. In this context, ‘You Are Not Alone for China’ could be regarded as a substitute for not fulfilling them. So, applying Bronfenbrenner’s bioecological model of human development [28], we may consider that chronosystem is not only the most critical aspect in this pandemic but also the one that unifies and interconnects the other levels of interactions. Through a societal-ecological model, the analysis shows the key role of chronosystem, the fine-tuning and melting of individual-micro-meso-exo and macro-systems in this matter of public health. This fact is also shown in the lyrics’ adaptation, which integrates images that reflect these different socio-ecological levels. In contrast, the original song mainly was focused on a man and his memories, with Michael Jackson and his wife, Lisa Marie Presley, reproducing the oil scene onboard Daybreak.

### 4.3. Rituals and Collective Mourning

From a societal-ecologic perspective, the original version goes through mourning phases restricted to the individual (the lonely man) and microsystem level (the voice that whispers). In contrast, the version for China also involves all the other mesosystem, exosystem, and macrosystem, as the song’s purpose is to express support to their second home. There is a jump from an individual scenario (original) into an exosystem (version), from the individual to the societal mourning. As discussed before, the chronosystem’s role is also important, as it intensifies the severity of the new described situation. It is known that the responses to shocking events are subjective and largely depend on our age, sensitivity, the interpersonal relationship with the person deceased, the social environment and culture, among other complex individual and socio-demographic factors [8,9,46]

A cultural perspective of the sources of trauma, at the individual and social level, and comparison from previous viral outbreaks, points to the current pandemic’s striking social dimension [7]. Under the gender perspective, sex differences have been found in the biobehavioral responses to threat and stress [29]. The fight-to-flight responses, presented by classical psychology as one model that fits it all, are now being reconsidered by evolutionary psychology. Taylor’s ‘Tend-and-Befriend’ theory [29] proposes affiliative behaviors and tending activities of females to females of different species, including humans, reducing biological stress responses in both parents and offspring. Thus, ‘Befriending’ (affiliative and peer support) would reduce stress-related health threats instead of the deleterious effects of social isolation. ‘Tend and Befriend’ has also been proposed as an intervention for nurses’ moral distress in intensive care units [47]. Here, we could propose ‘Tend-and-Befriend’ as a paradigm for many social expressions observed in this COVID-19 pandemic, including those related to social or collective mourning. Social bonding and group cohesion have been prompted to counteract compulsory social isolation due to physical distancing policies to hamper the COVID-19 pandemic.

According to the mourning process phases, the original video mainly focuses on the initial shock and denial. It has some bits of anger and bargaining and a jump to acceptance. In the version, the expressions of shock and denial of the first stanza are already transformed into positive thinking. The central theme is mainly focused on bargaining, messages of encouragement, support, and feeling together despite severe confinement. Regarding the dual process model of coping with grief [31], the versioned song, more than the original, could also be understood as disclosing the search for a healthy balance between the grieving state (loss-oriented stressors) and the coming life events (focusing on new tasks and life changes). This interpretation would agree with what many authors also found in their songwriting interventions [24]. Moving between these two types of behaviors is regarded as healthy coping, whereas focusing exclusively on the loss-oriented stressors would signify complicated grieving. This dual process model is gaining acceptance within the bereavement field and is particularly pertinent and applicable to the current study.

As mentioned before, collective mourning can be understood as an expression of societal maturity, cohesion, and respect [14]. During the first year of the pandemic, few countries paid public tribute to the victims of COVID-19 in a state ceremony. Most communication media also omitted death and signs of grief [48] but enhanced messages, calls to action, concerning confinement, first, followed by physical distancing after. Thus, there was an apparent social taboo of death. Singularly, in Spain, a national and international institutional representation, including the World Health Organization’s director and relatives of the deceased, attended a civil ceremony held at the Royal Palace of Madrid in July 2020 and transmitted online and through television. Universal rituals were done, and symbolism was used, such as the deposition of white roses next to a cauldron lit to remember those killed by the pandemic and reading a poem. The act was ended with a minute of silence. Similarly, at small locations (i.e., El Masnou - Barcelona), the purity of white roses and music performed by the youngest chamber orchestra were universal rituals and symbols used to hold a unifying ceremony tribute to the victims (mostly older adults) and to support to their grieving families.

### 4.4. A Ballad, a Song That Tells a Story

Ballads are songs that tell a story. Their romantic themes usually aim to celebrate past times, give significance to the present day, or contribute to building a better future. Human neuroscience has demonstrated such an extraordinary capacity not only to evoke feelings and recall past events but to contribute to emotional regulation and enhancement of empathy [17]. An American healthcare organization [49] has even used the ballad’s concept as a name because of its meaning. It poetically describes this particular choice based on a description of what a ballad is and means, which would also apply here in the interpretative analysis of ‘You are not alone for China’. The organization refers to ballad choice because it paints a picture of an event or a person, helping to find connections and understanding elements. Here, the ballad helps to express support and sympathy to the grieving Chinese community. Its lyrics can tell a story where the coexistence of love and loss enhances grief. The slow tempo and groove of the ballad’s music also contribute. Other issues such as tonality, modality, and harmonic planes, which are essential to understand the effect of music on people [17], would also determine why the ‘You’re Not Alone’ ballad was chosen to be the musical vehicle to describe the feelings of loss of a man and his thoughts, and now, the same features drive the adapted songwriting and imaginary describing feelings and thoughts of the authors of the version for China.

Love and war are the two main themes of traditional Chinese songs, and they are found melted in this adapted version (here, would be love and the virus). In fact, “You Are Not Alone” is an R & B & Electronic Music ballad about love and isolation, written by Roger S. Kelly after the loss of a close person, based on the theme “If We Can Start All Over” originally composed in 1993 by Belgians Danny and Eddy Van Passel. The title of that original theme already referred to resilience and was presented in plural form.

‘You are not alone for China’ could be contemplated as an artistic expression of grief with some aspects of anticipated social mourning. Apart from cross-cultural beliefs since early history considering music to have a healing effect on mind and body [50], musicologists and psychologists have studied its effects on physical, psychological, and social well-being. They refer to musical experiences at farewell ceremonies as important to give the bereaved a sense of meaning, hope, and courage [51]. Similarly, experiences of the grief process related to music choice demonstrate the consolidation role of music in farewell rituals and the mourning process [14]. The relatives’ role in farewell music and lyrics selection shows the association of songs to positive memories, feelings of belonging, and identification [52]. Besides, listening to self-selected music increases well-being more than that chosen by others [52]. The ballad’s choice and the elaboration and performance of the music video ‘You are not alone for China’ helped the participants in their grief, as confirmed by their report at ‘*Expat musicians present heart-touching music video to support China*’ published in Shenzhen Daily, a local (yet, of a 12.5 million people city) report [35].

### 4.5. The Imagery of the Crisis in Wuhan as Part of the Storytelling

The imagery integrated into the version strongly described the impact of the crisis and its vast dimensions. Decision fatigue in healthcare professionals, traumatic grief, bereavement in people who lost loved ones, loss of roles and identity, and social divisions related to the economic shutdown have been identified as potential sources of trauma [7,13]. Previous experiences, current events, and compassion fatigue are also a triad that triggers physical and mental health problems in less resistant or resilient individuals [53]. Still, most critical opinions warn that in such an unprecedented COVID-19 scenario, the urgent need to use ready-made models—that are impossible to fit it all—demands discussions with a broader vision towards the future [14,54]. Here, one could also be aware of the sudden derangement of the world and the socio-ecological complexity of the current scenarios. Voices also warn about the actual risk of stigmatization not only individuals but also countries, neglected groups, and priorities, underestimating individual and social resilience that results in most citizens being unable to cope [9,12,14,55]. In this respect, the adapted lyrics and the imagery do strongly rely on the purpose of showing resistance and resilience: they present strong empathy and social cohesion, recognition of the severe situation, and the vast physical, emotional and technical efforts of healthcare professionals and other frontline workers. They express heartfelt support and sympathy for bereaved families. The spread of this music video through social networks agrees with the use of social networks as a valuable source of support and imagery, encouraging narration and sharing, as a powerful tool to facilitate the grief process [56].

### 4.6. Lyrics and Songwriting

Songwriting plays a central role in music therapy. Most songwriting analysis methods allow the investigation of the songwriter’s self-concept, attributes, and aptitudes. However, the delivery largely depends on the methodological or theoretical perspective. Several research works have shown and discussed [25,57] that experiential approaches, like the one used in the present work, are considered more suitable when ambiguous and complex content is to be explored. They unveil psychological and developmental processes that may be missed by analytical deductive analysis focused on the themes [57].

In agreement with recent work [25], the content analysis revealed that the versioned song was explicit, including togetherness. The imagery included in the music video safely expressed grief-related emotions. Experiences were communicated verbally and non-verbally, as well as coping skills and opportunities to stay stronger. Therefore, participants’ engagement in this version of ‘You are not alone for China’ can also be understood as a spontaneous, not-intended collaborative therapeutic songwriting with grieving peers. In agreement with a resilient songwriting program for adolescent bereavement [25], this songwriting might decrease grief levels, enhance creative expression, and provide social support to their adoptive home. As discussed for music, songwriting could have also positively affected themselves and their international community. The words about ‘standing by’ or ‘staying’ pose the singers/songwriter’s attitudes in an affiliative and supportive position towards those in need. Before submitting this manuscript, a journalist reported about the version ‘You Are Not Alone for China’. Its authors confirmed most of the above interpretations concerning the aims and their experiences [35].

### 4.7. Fast Frame Intervals and Close-Up Shots

The length of shots affects a film and the visual communication value of frames. Scenes were edited for the observer to see through the eyes or reach the storyteller’s understanding. There was a clear choice of close-up shots and shorter duration in the current version, two tolls transforming the goals pursued. While the longer duration of shots can make a scene appear slower-paced or relaxed, shorter duration results in scenes that are more likely to be interpreted as pressing and faster-paced. Thus, The shorter duration contributed to express (by the actors) and perceive (by the observers) the fast and exponential temporal dimension of such unexpected scenario. Frames with medium shots where lead singers (foster sons/daughters) stand static using only gestures, alternated with frames illustrating machinery’s fast activity to build a hospital and people hugging. In the original version, the different choice of frame intervals and shots helps to present/perceive a man feeling lonely, in the middle of crowds or empty scenarios, in an unwanted and uncomprehended situation that remains static without improvement, as reinforced by the lyrics of ‘another day has gone, I’m still alone’.

Facial expressions and eye contact have a key role in non-verbal communication. The increase of close-up shots in the version for China emphasized these emotional aspects and promoted sympathy and empathy. A clear example was using the high-angle shot on a healthcare worker in a crouched position talking to someone by videoconference. Then the ‘over the shoulder shot’ let us know she was touching her baby through the mobile screen before she started crying. Close-up and Italian shots reinforced the eye-camera angle focusing on the eyes and expressions of physical fatigue. They expressed the need to share and/or warn about the severity of the emotional and physical pain and sorrow and the societal support and resilience.

### 4.8. We Are the World, We Are the Children

The music video ‘You’re not alone for China’ offers a heterogeneous but unified representation of the world through the structure, the number of photograms, gender composition, and racial diversity. The presence of native children at the end of the video could be understood as looking to the future. Also, as a sign of anti-ageism, all the age groups support older people. This is important in oriental societies’ gerontocracy, where the older adults’ social status is innate, as shown by their language. Etymologically, the concept of aging is part of socially respected words such as professor, masterwork, and homeland, among others.

Cultural values and beliefs are important in anticipatory grief being determinant when improving communication with those confronting grief and their families [55]. Here, it is also important to note that WHO is warning that as coronavirus spreads, xenophobia and anti-Asian racism have been reported [54,58,59]. Research works such as ‘Fighting against the common enemy of COVID-19: a practice of building a community with a shared future for mankind’ are illustrative [54]. As mentioned before, the ‘befriending’ under stress [29] would also apply in this analysis, as the gender roles are presented in a dichotomist manner. Females are the lead singers that transmit the message; males and one female are playing the music. Only in the final choruses mixed gender and group representations are shown.

### 4.9. Music and Social Convergence in Times of Physical Distancing

The video started with a car entering an empty three-lane highway. It is interesting to note the convergence on the emotional expression of loneliness used in the musical entrance, which is purely instrumental. However, the contra posed way used in both versions to illustrate it: feeling lonely amidst the social crows (original) vs. a single car in an absolutely empty Chinese transport system (version). As it would be reported later worldwide and be covered in magazines such as National Geographic, the singularity of empty cities and transportation systems presented here was more dramatic. It was the first time of such a situation and Shenzhen, considered the Silicon valley in Asia, is a city of 12.53 million inhabitants. The central theme of the ballad is loneliness.

In the original version, the lyrics and images explicitly declare the physical distancing and the chance to say goodbye. The encouragement message arrives in the chorus, a ‘You’re not alone’ to mitigate physical solitude with emotional empathy and support. In both the original and the version for China, these aspects are present for the individual and the collective loneliness, respectively.

Because of the effect of isolation on mental health, the World Health Organization warned about the inappropriate utilization of the term’ social distancing’ to allude to the measures to increment physical space between individuals to forestall the spread of COVID-19 [58]. It likewise features the importance of referring to ‘physical distancing’ instead. WHO reminds us that it is important to avoid disengagement from family and friends because, in the current situation, there is a need to remain genuinely and socially connected [60]. Most of the adapted lyrics of ‘You are not alone for China’ anticipated these aspects and concerns.

Social belonging, compassion, and kindness have been described as key ingredients for fostering resilience, recovery, and growth from the COVID-19 pandemic [60]. In this scenario, the role of music is evident, especially concerning its value for social convergence in physical distancing times. Music-related aspects of health and well-being are under study for their scientific interest in cultural psychology, and soon, it will also provide evidence of its role in social convergence in this pandemic [61]. All of these will talk in favor of the cultural sector that is now drowning in the socio-economic crisis.

The universality of music made the first journalist and television reports of the Italians under quarantine singing and playing music on balconies to go around the world [62]. The heartfelt compassion of a violin concert by Lena Yokoyama from the roof of a hospital in Cremona, the violins’ city, served as a tribute to health workers fighting the devastating pandemic scenario in Lombardia, the focus of COVID-19 in Italy. A piece of the music she performed, The Mission (Gabriel’s Oboe) by Ennio Morricone, evoked emotions and empathy worldwide. Similarly, years ago, a mystery man brought his piano to the Bataclan concert hall to play John Lennon’s ‘Imagine’ as a tribute to those affected by the tragedy in Paris.

In Spain, a call to action to stay at home (#quedateencasa, #stayathome) was promoted with song performances of the popularly called ‘quarantines’, such as ‘Quedate en Casa’ (Stay at Home), with great success among families with children and collectives. Rituals for social convergence in neighborhoods fast spread. While the Italians sang the national anthem at 6:00 p.m. in a call organized on social media every day, Spaniards applauded all frontline workers two hours later, at 8:00, in the so-called ‘Aplauso sanitario’ (Health applause). In all cities and towns, this sincere tribute was followed by performances, from opera music to magic, from balconies. Among them, Pau Dones, lead singer of famous Spanish group Jarabe de Palo, who wholeheartedly dedicated the song ‘Los ángeles visten de blanco’ (The angels wear white) to frontline workers a few months before passing away. Precisely at the same hour, every evening until May 11h, music melted with sounds of artificial respirators of intensive care units and lights symbolizing the souls going to heaven transformed the Eiffel Tower into a tribute and call to stay at home [63]. In September, 2020, Barcelona’s ‘piromusical’, a major musical firework spectacle as the closing ceremony of its local festivity ‘La Mercè’ was closed to the public but complemented with other simultaneous musical firework showed at different districts and broadcasted live on tv with a vintage soundtrack that paid homage to Barcelona’s oldest citizens.

Memorable songs, internationally recognized and awarded, have been re-edited. The Michael Jackson Foundation [64] re-edited a masterpiece: ‘Heal the World, 2020’. In its description on YouTube, the Foundation mentions the capacity of music performances and music shared online to unite the world during the COVID-19 pandemic. The new version used Michael Jackson’s live performance melted with imagery of the current crisis to send a message of support and comfort and the wish that Michael Jackson’s music will continue to unite the world.

In Spain, ‘Resistiré’ (I will resist), a famous pop song from 1988 ‘Dúo Dinámico/Dynamic Duo’ formed by Manuel de la Calva and Ramón Arcusa, was re-edited as ‘Resistiré 2020’ and performed at home by different artists. This musical duo was very popular in Spain in the 1960s, so the song was especially meaningful for the oldest population. The frontline workers at care units, and several other collectives, even the police force, performed this song and shared their versions online to encourage the oldest people, the children, and the general population. It is also important to note that this song is among those with a high groove that are the leitmotiv of cancer patients and survivors, as are ‘I Will Survive’ by Gloria Gaynor, or ‘You’ve Got a Friend’ by Carole King. The emblematic Barcelona’s 92 Olympic Games farewell song ‘Amigos para Siempre’ (Friends forever) was also versioned by Los Manolos to recognize and support frontline workers. The top 10 lists of songs inspired in COVID-19 have been edited and include famous musicians such as Bono, Alejandro Sanz, Jorde Drexler, and Neil Diamond. They sang about the pandemic with new and versioned songs. Finally, the global broadcast, digital special ‘One world together at home’ show organized by Lady Gaga, Global Citizen, and the campaign rallying funds to support frontline healthcare workers and the WHO, are probably the most significant cultural demonstration of the current cohesion worldwide building resilience [64]. These great songs that authors and society have spontaneously chosen to be performed alone or together to cope with the sudden and dramatic situations of this COVID-19 pandemic, mitigate physical distancing, and alleviate human suffering, as previously discussed. Thus, despite the wide variety of cultural and traditional rituals related to death-centered music, the current artistic expressions demonstrate that music is an international cross-cultural common language.

## 5. Conclusions

‘You are not alone for China’ music video, where music is blended with COVID-19 adapted lyrics and integrated with shocking imagery describing the outbreak in Wuhan, has been analyzed as a spontaneous expression of copying with such traumatic events and mourning process at the individual and collective level. The analysis depicts several of the known elements such as the music, adaptive lyrics, group songwriting—signing and playing music, cross-cultural elements and social cohesion that support and promote healthy grieving, providing support to music and songs as spontaneous expressions of human individual and social resistance and resilience in front of critical events. Through a societal-ecological model, the analysis shows the key role of chronosystem and the extraordinary melting of individual-micro-meso-exo and macro-systems in this matter of public health. The five stages of the mourning process described by psychiatrist Elisabeth Kübler-Ross were identified and compared between both versions providing a better understanding of the transition from interpersonal to societal mourning. A tend-to-befriend approach to express deep sorrow but yet provide faith and strength to the people of China. Finally, other extraordinary songs that authors and society have spontaneously chosen to perform alone or together to cope with sudden and dramatic situations, mitigate physical distancing, and alleviate human suffering are discussed. They have played a key role in building extraordinary social and emotional ties, which are among the best strategies for our individual and social pain and sorrow during this pandemic, breaking down cultural barriers. Music, lyrics, and artistic performances can be foreseen as a social balsam that will continue to play a significant psychological and social role.

## Figures and Tables

**Figure 1 behavsci-12-00088-f001:**
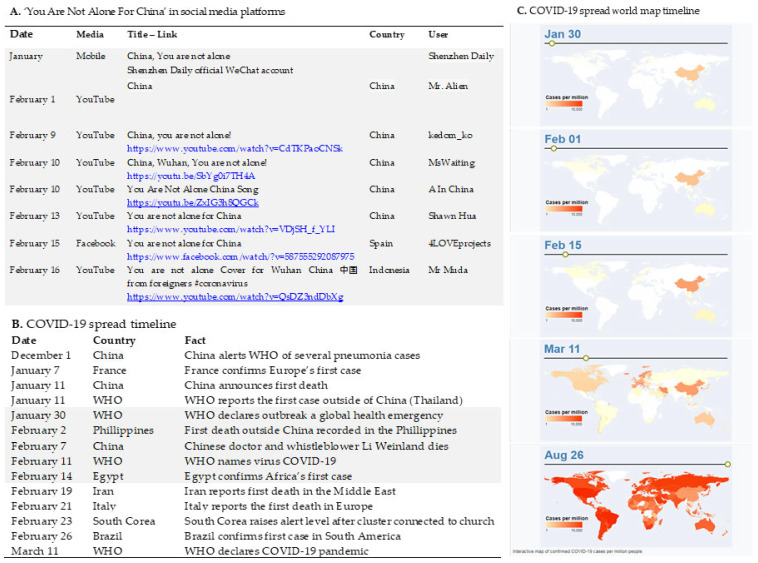
‘You Are Not Alone China in social media, COVID-19 timelines, and world map. ‘You Are Not Alone For China’ in Shenzhen Daily official WeChat account, and the timeline of the version as recorded when it reached open-access international accounts (**A**), COVID-19 spread timeline with grey color indicating the period corresponding to the appearance of the cover version (**B**), and COVID-19 spread world map timeline from its beginning to the worldwide dissemination (**C**).

**Figure 2 behavsci-12-00088-f002:**
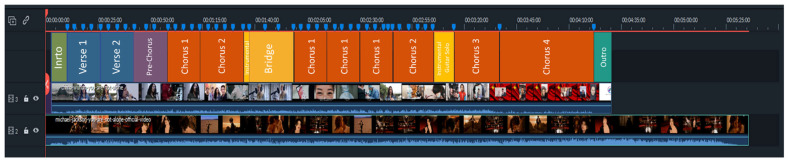
The musical structure of the cover version is compared to the original song. Rhyme scheme (**top**) of the cover version and video/audio length and composition of the cover (**middle**) and the original song (**bottom**). See also Figure 3, Figure 4 and Figure 5.

**Figure 3 behavsci-12-00088-f003:**
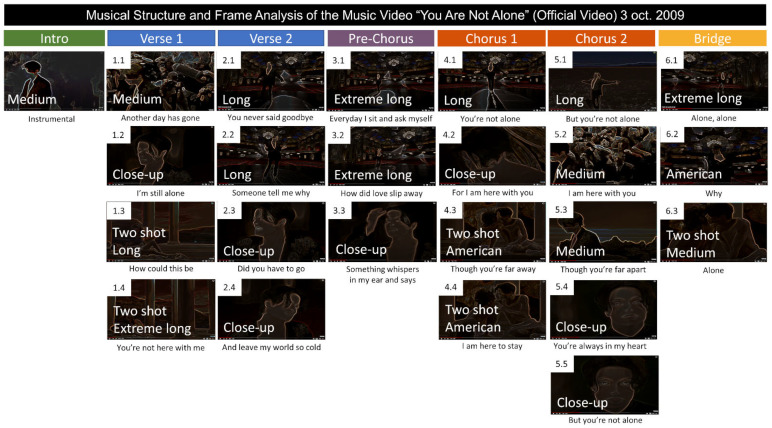
Musical structure and frame analysis of the music video ‘You Are Not Alone’ by Michael Jackson.

**Figure 4 behavsci-12-00088-f004:**
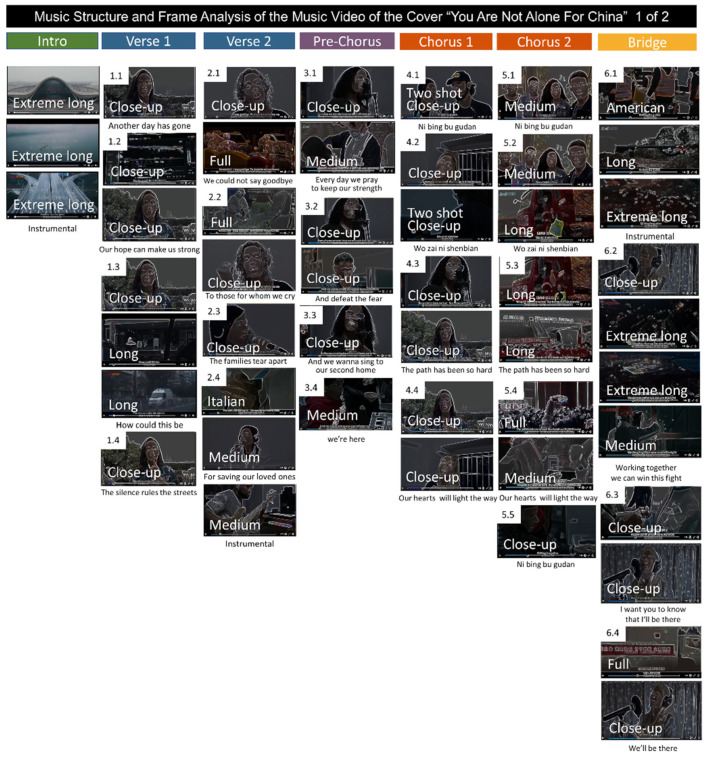
Musical structure and frame analysis of the music video of the cover ‘You Are Not Alone For China’ by young singers and musicians during the COVID-19 outbreak in China, in the parts in common with the original song.

**Figure 5 behavsci-12-00088-f005:**
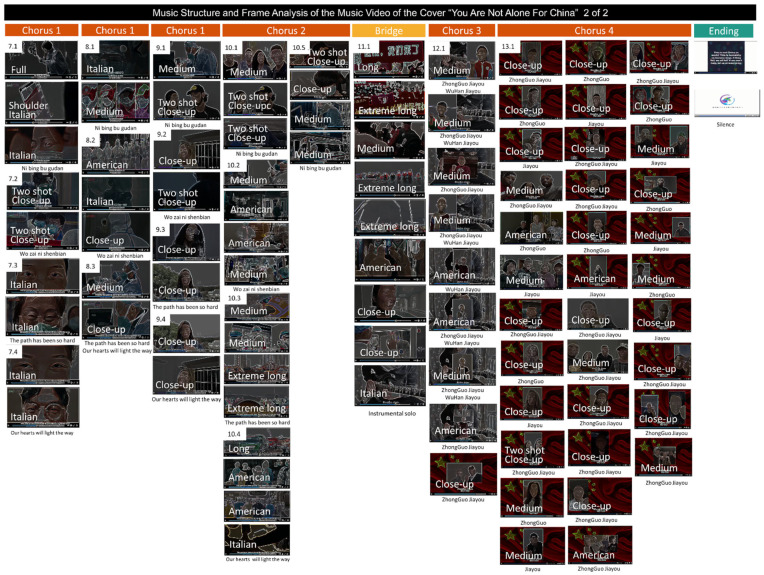
Musical structure and frame analysis of the music video of the cover ‘You Are Not Alone For China’ by young singers and musicians during the COVID-19 outbreak in China, in the extended parts.

**Figure 6 behavsci-12-00088-f006:**
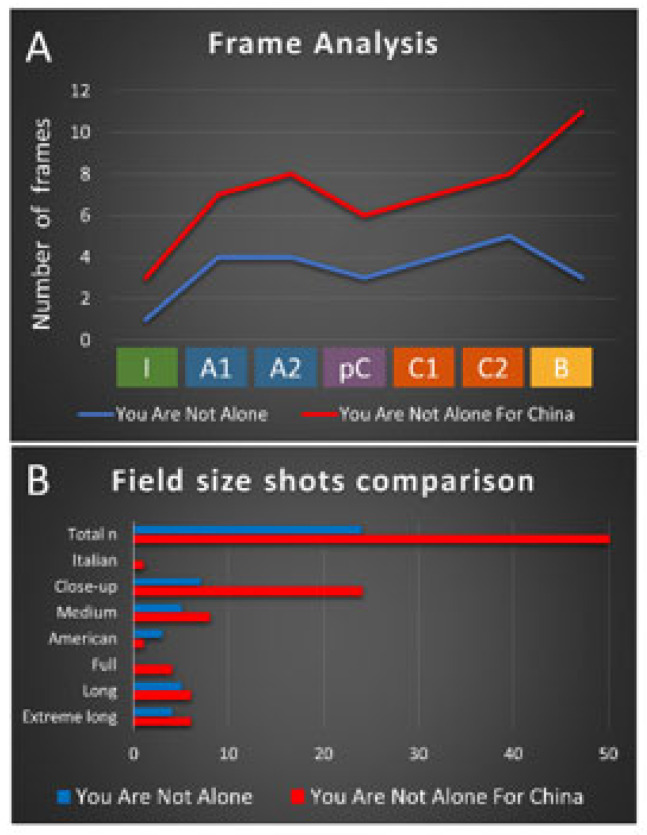
Frame analysis (**A**) and Field size shots comparison (**B**) between the original and the cover song.

**Table 1 behavsci-12-00088-t001:** Comparative analysis of the infobox song of ‘You are not alone’ and the cover version for China.

	Original Song	Cover Version for China
**Name**	You Are Not Alone	You Are Not Alone For China
**Artist**	Michael Jackson	Shenzhen Meifeiya Culture Communication Co. Ltd.
**Album**	HIStory: Past, Present, and Future, Book I	-
**Format**	CD single × Cassette, 7″ × 12″ × Video Promo	Music Video
**Language**	English	English and Chinese
**Country**	USA	China
**Recorded**	November, 1994	January 2020
**Released**	August 15,1995	January/February 2020
**Genre**	Pop × R&B	Pop × R&B
**Length**	5:45 (Album Version)	-
	4:34 (Radio Edit)	-
	4:56 (Single Version)	-
	5:35 (Video Version)	4:27 (Video Version)
	6:01 (Extended Version)	-
**Label**	Epic	-
**Songwriter**	Robert Kelly	Shenzhen Meifeiya Culture Communication Co. Ltd.
**Producer**	Robert Kelly and Michael Jackson	Shenzhen Meifeiya Culture Communication Co. Ltd.

**Table 2 behavsci-12-00088-t002:** Comparative analysis of ‘You Are Not Alone for China’ and the original song ‘You Are Not Alone’.

STRUCTURE	VERSE AND TEMPORAL FRAME		ADAPTATIONAND [PARTICIPANTS PERFORMING]	GRAMMATICAL AND FUNCTIONAL ANALYSIS	
**VERSE.STANZA**	**You Are Not Alone.** **Original Song**	**You Are Not Alone For China.** **Version COVID-19**		**You Are Not Alone.** **Original Song**	**You Are Not Alone For China. Version COVID-19**
**VERSE 1**	**Another day has gone** 0:10–0:13	**Another day has gone** 0:05–0:10	None (Preserved) [Woman 1]	Pass of time empty of her love	Pass of time empty of her love
**1.2**	I’m still alone 0:13–0:16	Our hope can make us strong 0:10–0:16	Changed to positive Present into Future [Woman 1]	Persistence of Loneliness	Hope as a source of strength
**1.3**	**How could this be** 0:16–0:21	**How could this be** 0:16–0:19	None (Preserved) [Woman 1]	Incomprehension Incomprehension	
**1.4**	You’re not here with me 0:21–0:25	The silence rules the streets 0.19–0:24	Homologous [Woman 1]	Confirmation of absence Present	Confirmation of absence of people in the streets Present
**2.1**	You never **said goodbye** 0:25–0.30	We couldn’t **say goodbye** 0.24–0:28	Partially preserved but a change of meaning (adaptive) [Woman 2]	The other is guilty of not doing the act of saying goodbye	The group could not say goodbye
**2.2**	Someone tells me why 0:30–0:33	To those for whom we cry 0:28–0:31	Change of meaning [Woman 2]	Start of a new sentence. Incomprehension Request answer to others	Confirms there’s a pain and the situation was not chosen
**2.3**	Did you have to go 0:33–0:37	The families tear apart 0:31–0:35	Concept of Social distance is preserved, but with a different meaning [Woman 3]	The other person had a reason to go	The families are emotionally devastated by the separation
**2.4**	And leave my world so cold 0.37–0:42	For saving our loved ones 0:35–0:41	Change of meaning (adaptative) [Woman 3; Man 1]	Abandonement. The result is to leave the subject and his world with sadness.	Opposite meaning, to indicate commitment The reason or expected result is to save beloved ones.
**3.1**	**Every day** I sit and ask myself 0:42–0:47	**Every day** we pray to keep our strength 0:41–0:45	Time preserved But a change of meaning (adaptative) [Woman 4]	Loneliness and Sad intrusive thoughts	Attitude searching for hope, together
**3.2**	How did love slip away 0:47–0:50	And defeat the fear 0:45–0:50	Change of meaning (adaptative) [Woman 4]	Sense of loss and incomprehension	Recognition of fear Attitude towards success
**3.3**	Something whispers in my ear and says 0:50–0:57	And we wanna sing to our second home we’re here 0:50–0:55	Change of meaning (adaptative) [Woman 4; Men 2 and 3]	An unknown external source provides information	Recognition of affiliation Attitude of using internal resources to express support
**CHORUS**	** That you are not alone ** **0.57–1:02**	**Ni bing bu gudan** (You’re not alone) 0.56–0:59	Use of Chinese [Woman 3 and man 4]		
	** For I am here with you ** **1:02–1:06**	**Wo zai ni shenbian** (I’m standing by your side) 0:59–1:03	Use of Chinese [Men 2 and 3]		
	Though you’re far away 1:06–1:09	The path has been so hard 1:03–1:07	Change of meaning (adaptative) [Women 4 and 1]	Physical distance External attribution of loneliness	Reference to resistance. Overcome difficulty of a path
	I am here to stay 1:09–1:12	Our hearts will light the way 1:07–1:10	Change of meaning (adaptative) [Women 1 and 5]	Reference to external source of resistance	Reference to resilience, internal source
	**But you are not alone** **1:12–1:17**	**Ni bing bu gudan** **1:10–1:15**	Use of Chinese [Men 5 and 6, woman 6 and man 7]	Reference to external source to confront loneliness	Reference to external source to confront loneliness
	**I am here with you** **1:17–1:21**	**Wo zai ni shenbian** **1:15–1:19**	Use of Chinese [Men 5 and 6, woman 6 and man 7]	Reference to external source of support	Reference to external source of support
	Though we’re far apart 1:21–1:24	The path has been so hard 1:19–1:23	Change of meaning (adaptative) [All, indeterminate]	Physical distance External attribution of loneliness	Reference to resistance. Overcome difficulty of a path
	You’re always in my heart 1:24–1:29	Our hearts will light the way 1.23–1:27	Partially preserved [All, indeterminate]	The heart as a source of the immortality of LOVE	The heart as a source of the light to continue (resilience)
	**You are not alone** **1:29–1.3**	**Ni bing bu gudan** **1:27–1:32**	Use of Chinese [Men 2 and 3]		
	Alone, alone 1:33–1:39 1:39–1:42	(Music) 1:32–1:34			
	Why, alone 1:42–1:44 1:44–1:50	Working together we can win this fight	[Woman 7]		Recognition of affiliation and social cohesion Attitude towards success
		I want you to know that I’ll be there	[Woman 7]		Expression of individual support
		We’ll be there	[Woman 7]		Confirmation of social support
		Ni bing bu gudan Wo zai ni shenbianThe path has been so hard. Our hearts will light the way (4x)	[All, indetermined] [Woman 3 and man 4; Woman 5; Men 2 and 3; Woman 4; Woman 1; Woman 5; Men 5 and 6, woman 6 and man 7; Woman 3 and man 4; all indetermined]		
		Ni bing bu gudan			
	3:11-end	ZhongGuo Jiayou (China, stay strong!)	[Girl 1; boy 1; girl 2; boy 2; girl 3; boy 3; girl 4; girl 5; Woman 7 and man 8; Men 9–39; Women 8–23]		
		Wu Han JiaYou! (Wu Han, stay strong!)			
		ZhongGuo Jiayou!			

**Table 3 behavsci-12-00088-t003:** Comparative analysis according to Psychological First Aids.

Comparative Analysis According to Basic Aspects in Psychological First Aid		
	You are not alone	You are not alone for China
Starring actor	Man	Foreigners (Foster son/daughters)
Directed to	Woman	People of China and Focus of the Epidemic
Goals	Showing loneliness and asking for answers	Offering support and providing answers
Facts	Loss of a woman’s love	Confronting COVID-19 lockdown and loss of beloved, while the rest of the world is yet not affected
Cognition	Incomprehension Asking answers External individual resources to change the situation	Incomprehension Asking answers Pointing at internal collective resources to change the situation Social cohesion and affiliative thinking
Emotions	Surprise/Anger/Sadness	Surprise/Negation/Sadness/Bargaining
Actions	Freezing	Active Express support Social cohesion and affiliative behaviors
Time	Past, present	Past, present, and future

## Data Availability

The data presented in this study are depicted in Figure 6.

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
