# Peer review of "‘You’re Not Alone for China’: The First Song in Times of COVID-19 to Keep the Faith in a World Crying in Silence"

_behavsci, 2022, doi:10.3390/bs12040088_

Round 1

Reviewer 1 Report

The article presented is of interest, however, throughout the theoretical framework the authors systematically repeat the importance of music and its impact on the evocation of memories, the capacity for empathy and the power it has. in the feelings of the individual which causes that bond of union to exist in the collective mourning. It is curious that the authors analyze only non-verbal issues, images, lyrics… but at no time is there any analysis of music that would be essential to understand precisely the study variables being worked on, as well as the effect on the population. . Issues such as tonality, modality and harmonic planes are essential to understand the effect of music on people who view the video which of course is accompanied by images that are also of vital importance.
If the analysis is performed only towards the letter or the image in this case the music reduces interest in the object of study and precisely focuses on the contrary the whole theoretical framework.
Perform a musical analysis or specify in this paragraph "The original song and its version were compared in terms of its musical structure and main features" which is the musical structure and the main characteristics of the music is not enough to analyze the structure as a musical level does not make a consistent contribution.

This is the fundamental problem that I consider in this article since there is no coherence between the development of the theoretical framework and the analysis carried out by the authors. Although the article is of sociological interest, I believe that the type of analysis and approach of the article should be reviewed.

Author Response

ANSWER TO REVIEWERS

First at all, I’d like to thank the referee for her/his time to review the submitted Ms. I appreciate that the referee finds the article presented of interest and I thank for the scientific discussion provided, through a number of questions and suggestions, to improve the quality of the work.

In the next paragraph I’d provide a point-by-point answer to her/his queries and I’ll indicate how the Ms has been modified according to them.

REVIEWER 1

The article presented is of interest, however, throughout the theoretical framework the authors systematically repeat the importance of music and its impact on the evocation of memories, the capacity for empathy and the power it has. in the feelings of the individual which causes that bond of union to exist in the collective mourning. It is curious that the authors analyze only non-verbal issues, images, lyrics… but at no time is there any analysis of music that would be essential to understand precisely the study variables being worked on, as well as the effect on the population. . Issues such as tonality, modality and harmonic planes are essential to understand the effect of music on people who view the video which of course is accompanied by images that are also of vital importance.

If the analysis is performed only towards the letter or the image in this case the music reduces interest in the object of study and precisely focuses on the contrary the whole theoretical framework.
Perform a musical analysis or specify in this paragraph "The original song and its version were compared in terms of its musical structure and main features" which is the musical structure and the main characteristics of the music is not enough to analyze the structure as a musical level does not make a consistent contribution.

This is the fundamental problem that I consider in this article since there is no coherence between the development of the theoretical framework and the analysis carried out by the authors. Although the article is of sociological interest, I believe that the type of analysis and approach of the article should be reviewed.

The author of this Ms completely agrees with the referee’s consideration on the relevance to study variables such as tonality, modality and harmonic planes as essential to understand the effect of music on people. In agreement, here, one could analyze the intrinsic value of these study variables in the original ballad ‘You’re not alone’ to achieve the goal of the songwriter to express the feelings of loss of a lover and his thoughts. The same music, the ‘You’re not alone’ ballad, was used in the new version for China, where the authors transformed the song by shortening the length of the ballad (musical structure), but mostly by changing the lyrics and images to describe their collective loss and to express their grief. In this respect, to the music per se, in the discussion there is a subsection entitled A ballad, a song that tells a story lines 276-310, and also in subsection Lyrics and songwriting lines 349-361 where it is discussed that “Songwriting plays a central role in music therapy”, but definitely it is not the focus of the present work and I apologize if the body text drives to that expectation. Therefore, as author of this Ms. my interest was not to analyze the You’re not alone ballad’s musical features and how they are able to drive emotions and feelings in the original, and also how these same features are also the same vehicle in the cover version. Certainly, that would be of great interest, but that would need a more extended Ms or a final Ms different of the current one. Please, note that as it is mentioned in the introduction and the aims of the study, the analysis is focused on the use of a specific ballad (You’re not alone) by young foreigners living in China to express individual and collective sorrow in the context of the outbreak. The Ms is aimed to describe the modifications that the authors of the cover version for China did to adapt the ‘lyrics’ of the original song to express their feelings and thoughts. They did not modify the original song, but shortened the structure (as shown by the results), whereas the main modification was observed in the songwriting. In fact, the analysis of such ‘adaptative’ process observed from one to the second version, is the base of several artistic interventions where the person in grief is given a ‘material’ with the expectation that, in the process of transformation to the new version (her/his version), the person is able to express emotions and thoughts, and elaborate her/his grief at the individual or at the group level. In my understanding such an analysis of the ‘songwriting adaptation’ and other analysis performed is a contribution not just on sociology but psychology of grief. In agreement with the above, the text has been revised to be more specific on the purpose of the current work. Changes in the text have been highlight in red to be easier to track.

I hope the referee will find the modifications appropriate to clarify this issue and drive the attention to the songwriting process (as the main target) and other adaptations, as it is indicated in the abstract: “The song ‘You Are Not Alone’ was adapted as a cover version by young foreigners living in China to express their support to bereaved families and frontline workers and encourage the people of China, their second home”. Also, another very explicit paragraph was  “Music and songs can also provide scenarios and tools for individual and group interventions [20,21]. Lyric analysis, improvisation, singing, and songwriting from bereaved children, preadolescents and adolescents [22-24] are used in resilient programs tailored for individuals and as part of group music therapy [25]. Thematic analysis through mixed inductive and deductive content analysis methods is also used [26,27]. Probably, this is the one that better introduces the purpose of the current study.

Sincerely yours,

Lydia Giménez-Llort

Reviewer 2 Report

Thank you for resubmitting this interesting paper.

Author Response

Thank you for your time to review the resubmitted version that included all your constructive suggestions and comments, and for providing a final positive consideration.

Sincerely,
Lydia Giménez Llort

Reviewer 3 Report

I find all of my previous comments having been correctly  tacked with

Author Response

Thank you for your time to review the resubmitted version that included all your constructive questions, suggestions and comments, and for providing a final positive consideration.

Sincerely,
Lydia Giménez Llort

Round 2

Reviewer 1 Report

Dear editor,

Regarding the article that I have corrected, I have serious doubts that once some of the proposed modifications have been made, but not the one that for me is the main one (the analysis of the music of the video that they analyze only focusing on the visual part), it could be interest for publication. If I take into account the revisions carried out, I could accept it, however I consider that the object of study is in disagreement with the analysis carried out or I believe in my opinion that it is incoherent to speak of music and music therapy when a technical description of this variable is not made.